# Estimating $R_e$ and overdispersion in secondary cases from the size of identical sequence clusters of SARS-CoV-2

Emma B. Hodcroft [1,2,3☉¤*], Martin S. Wohlfender [1,2,4☉], Richard A. Neher [3,5], Julien Riou [1,2,6], Christian L. Althaus [1,2]

**1** Institute of Social and Preventive Medicine, University of Bern, Bern, Switzerland, **2** Multidisciplinary Center for Infectious Diseases, University of Bern, Bern, Switzerland, **3** Swiss Institute of Bioinformatics, Lausanne, Switzerland, **4** Graduate School for Cellular and Biomedical Sciences, University of Bern, Bern, Switzerland, **5** Biozentrum, University of Basel, Basel, Switzerland, **6** Department of Epidemiology and Health Systems, Unisanté, Center for Primary Care and Public Health & University of Lausanne, Lausanne, Switzerland

☉ These authors contributed equally to this work.
¤ Current Address: Swiss Tropical and Public Health Institute, Allschwil, Switzerland
* emma.hodcroft@unibas.ch

**Data availability statement:** Sequence data are available via GISAID after registration, and are available in EPI_SET_240326pm

## Abstract

The wealth of genomic data that was generated during the COVID-19 pandemic provides an exceptional opportunity to obtain information on the transmission of SARS-CoV-2. Specifically, there is great interest to better understand how the effective reproduction number $R_e$ and the overdispersion of secondary cases, which can be quantified by the negative binomial dispersion parameter $k$, changed over time and across regions and viral variants. The aim of our study was to develop a Bayesian framework to infer $R_e$ and $k$ from viral sequence data. First, we developed a mathematical model for the distribution of the size of identical sequence clusters, in which we integrated viral transmission, the mutation rate of the virus, and incomplete case-detection. Second, we implemented this model within a Bayesian inference framework, allowing the estimation of $R_e$ and $k$ from genomic data only. We validated this model in a simulation study. Third, we identified clusters of identical sequences in all SARS-CoV-2 sequences in 2021 from Switzerland, Denmark, and Germany that were available on GISAID. We obtained monthly estimates of the posterior distribution of $R_e$ and $k$, with the resulting $R_e$ estimates slightly lower than estimates obtained by other methods, and $k$ comparable with previous results. We found comparatively higher estimates of $k$ in Denmark which suggests less opportunities for superspreading and more controlled transmission compared to the other countries in 2021. Our model included an estimation of the case detection and sampling probability, but the estimates obtained had large uncertainty, reflecting the difficulty of estimating these parameters simultaneously. Our study presents a novel method to infer information on the transmission of infectious diseases and its heterogeneity using genomic data. With increasing availability of sequences of pathogens in the future, we expect that our method has the potential to provide new insights into the transmission and the overdispersion in secondary cases of other pathogens.

(dx.doi.org/10.55876/gis8.240326pm) for Switzerland, EPI_SET_240326mz (dx.doi.org/10.55876/gis8.240326mz) for Denmark, and EPI_SET_240326uh (dx.doi.org/10.55876/gis8.240326uh) for Germany; see also S1-S3 Tables in the supporting information. All other data used in this study such as COVID-19 case counts come from public sources. The data files are contained in the GitHub repository github.com/mwohlfender/R_overdispersion_cluster_size. Their sources are cited in the readme file of the repository as well as in the manuscript or in the supporting information. Code to generate identical sequence clusters from the starting alignments is available via github.com/emmahodcroft/sc2_rk_public. Processed sequence data describing the size of the identified identical sequence clusters and the time period the sequences they contain were collected as well as code used for the analysis of data and results as well as the creation of plots and tables is available via github.com/mwohlfender/R_overdispersion_cluster_size. Functions for estimation of parameters and simulation of identical sequence clusters are available via the R package estRodis github.com/mwohlfender/estRodis.

**Funding:** This work was supported by the Swiss National Science Foundation (196046 to EH, RH, and CA), by the Swiss Federal Office of Public Health (189498 to JR) and by a Swiss National Science Foundation Starting Grant (TMSGI3_211225 to EH). This work was further supported by the Multidisciplinary Center for Infectious Diseases, University of Bern, Bern, Switzerland (to MW, JR, and CA) and the European Union's Horizon 2020 research and innovation program - project EpiPose (No 101003688 to CA). Additionally, this project was supported by the ESCAPE project, funded by the European Union (101095619 to CA and EH). Views and opinions expressed are however those of the author(s) only and do not necessarily reflect those of the European Union or European Health and Digital Executive Agency (HADEA). Neither the European Union nor the granting authority can be held responsible for them. This work has received funding from the Swiss State Secretariat for Education, Research and Innovation (SERI) (contract number 22.00482 to CA and EH). The funders had no role in study design, data collection and analysis, decision to publish, or preparation of the manuscript.

**Competing interests:** RAN has received consulting fees from ModernaTX and BioNTech in matters unrelated to this work.

## Author summary

Pathogen transmission is a stochastic process that can be characterized by two parameters: the effective reproduction number $R_e$ relates to the average number of secondary cases per infectious case in the current conditions of transmission and immunity, and the dispersion parameter $k$ captures the variability in the number of secondary cases. While $R_e$ can be estimated well from case data, $k$ is more difficult to quantify since detailed information about who infected whom is required. Here, we took advantage of the enormous number of sequences available of SARS-CoV-2 to identify clusters of identical sequences, providing indirect information about the size of transmission chains at different times in the pandemic, and thus about epidemic parameters. We then extended a previously defined method to estimate $R_e$, $k$, and the probability of detection from this sequence data. We validated our approach on simulated and real data from three countries, with our resulting estimates compatible with previous estimates. In a future with increased pathogen sequence availability, we believe this method will pave the way for the estimation of epidemic parameters in the absence of detailed contact tracing data.

## Introduction

The COVID-19 pandemic prompted an unprecedented global effort in generating and sharing SARS-CoV-2 sequences. As of March 2024, four years after the first sequences were released, over 16 million full-genome sequences have been shared, almost fifty times the estimated number of full-genome flu sequences generated over decades [1]. At the same time, scientists, governments, and public health authorities were keen to make use of all available methods to better understand the ongoing pandemic and implement appropriate responses. For example, information on the transmission dynamics of SARS-CoV-2, that can be characterized by the basic reproduction number $R_0$ and the effective reproduction number $R_e$, was critical to understanding whether current pandemic restrictions were controlling transmission [2] and whether new variants might cause pressure on healthcare systems. As well as estimating $R_e$ from the available case-count data [3,4], the availability of significant numbers of sequences created the opportunity to estimate this, and other parameters, through sequence- and phylogenetic-based methods [5–7].

Historically many of these techniques have been developed and applied to study pathogen dynamics in the past or in outbreak conditions when information such as reliable case counts may not be available [8–11]. However, another benefit of utilizing sequences is the ability to estimate parameters that otherwise require a level of detailed data acquisition that is difficult to obtain. A prime example of this is measuring the heterogeneity in transmission, which can be quantified by the dispersion parameter $k$. Estimating $k$ normally requires knowledge not only of the number of confirmed cases per day, but the number of secondary cases created by individual cases. In a pandemic or epidemic, $k$ quantifies how much spread may be driven by superspreading events, and thus can inform intervention efforts [12–17]. As part of contact-tracing efforts by many countries, estimates of $k$ using groups of linked cases have been possible, but due to the detailed level of data most require, have generally been limited to use on a few thousand cases and contacts [18–31], or simulated data [32]. In countries where SARS-CoV-2 cases were well-contained and thus had a high probability of being fully-traced, these datasets may be able to capture transmission dynamics well, but particularly as the pandemic has progressed and case numbers have risen, using relatively small subsets may both not fully capture the larger dynamics and may not be completely traced [33]. Using the large number

of readily-available SARS-CoV-2 sequences to estimate values such as $k$ could overcome these limitations.

Blumberg and Lloyd-Smith's [34] method provides a way to estimate $R_0$ (or $R_e$) and $k$ without requiring contact tracing data, and instead using a transmission chain size distribution. Their framework presumes transmission chains are finite with $R_e$ below 1, or in other words, the transmission chain must "stutter out." This method has been successfully applied to pathogens that spill over from animal reservoirs and don't lead to sustained human-to-human transmission, such as mpox prior to 2022 and MERS-CoV. In MERS-CoV outbreaks during 2013–2014, multiple spillover events from camels and well-traced transmission chains have resulted in datasets well-suited to estimating $R_0$ and $k$ [35–37]. However, even in instances such as these, where transmission networks are relatively well-recorded, cases may be missed, mis-attributed, or incorrectly considered a part of a transmission chain rather than a separate introduction due to insufficient background case detection. Thus, Blumberg and Lloyd-Smith [38] adapted their method to take into account imperfect case-detection.

During a pandemic, individual transmission chains might stop, but the state of being in a pandemic implies that overall these chains persist. Therefore, cases can not be readily decomposed into individual limited transmission chains. Instead, transmissions chains aggregate to macroscopic case numbers and the method of stuttering chains by Blumberg and Lloyd-Smith [34] can not be applied in such a scenario. To overcome this limitation, we propose to use clusters of identical sequences as a proxy for transmission chains: even in an ongoing transmission chain, the mutation rate of the virus means that every unique genotype will only exist until the virus mutates and thus is limited in size. To this end, we introduce a new metric, the genomic reproduction number $R_g$, that relates the reproduction number of cases with identical sequences to the effective reproduction number of cases, $R_e$. While concluding this work, Tran-Kiem and Bedford [39] published a similar study deriving the same mathematical model for the distribution of the size of identical sequence clusters based on the same theoretical concept. The development of the respective mathematical models was carried out in parallel and independently by both groups of authors, with considerable differences in the approach to estimating epidemiological parameters from genomic data, including an efficient way of determining genetic clusters, utilizing a Bayesian framework, and applying the method to much larger datasets.

In this study, we extended the method by Blumberg and Lloyd-Smith [34] and developed a Bayesian framework to infer $R_e$ and $k$ from viral sequence data. First, we developed a mathematical model of the size distribution of identical sequence clusters, in which we integrated viral transmission, the mutation rate of the virus, and incomplete case-detection to capture all aspects of the data-generating mechanism. Second, we implemented this model within a Bayesian inference framework, that we validated using simulated data. Third, we identified clusters of identical sequences in all SARS-CoV-2 sequences in 2021 from Switzerland, Denmark, and Germany that were available on GISAID and applied our method to obtain monthly estimates of $R_e$ and $k$ in these countries.

## Materials and methods

### Data

We downloaded all available SARS-CoV-2 sequences (14.8 million) from GISAID [1] on 7 March 2023. We ran them through the Nextstrain's *ncov-ingest* pipeline [40], which provides a list of all nucleotide mutations relative to the Wuhan-Hu-1/2019 reference (Genbank: MN908947) via Nextclade analysis [41]. Sequences identified as being problematic due to divergence issues or variant sequences reported prior to the variant's origin (dating

issues) were excluded. We then selected sequences from Switzerland, Denmark, and Germany, three countries with different testing and sequencing strategies resulting in different case-detection and sequencing coverage. The total number of sequences at the start of our analysis was 162,049 for Switzerland, 632,400 for Denmark, and 901,748 for Germany. All sequences are available from GISAID after registration as EPI_SET_240326pm for Switzerland, EPI_SET_240326mz for Denmark, and EPI_SET_240326uh for Germany (see also S1–S3 Tables).

To identify genetically identical sequences efficiently on a large number of sequences, the list of nucleotide mutations was used as a hash key, with sequences with identical hash keys being classified as a cluster. To minimize the chance that sequences with low numbers of mutations, which may have arisen independently, falsely cluster together, only sequences with more than four mutations, relative to the reference Wuhan strain, were used to form clusters. Due to the mutation rate of SARS-CoV-2, this limitation only excluded sequences at the very start of the pandemic and before our study period of 2021, as most lineages had more than four nucleotide mutations by March/April 2020.

Clustering by lists of nucleotide mutations is efficient but imperfect, as sequencing errors such as lack of coverage and use of ambiguous bases can impact whether sequence mutation lists hash identically. We assumed that such errors are randomly distributed through time and thus have minimal effect on the overall cluster size distribution of such a large number of sequences. It's infeasible to account for all possible sequencing errors, but in an effort to minimize their impact, for all recognized Nextstrain variants, variant-defining mutations as obtained from CoVariants.org [42] were removed and replaced with the variant name. This prevents variant sequences being erroneously separated if they are missing variant-defining mutations. The code used for this processing is available at github.com/emmahodcroft/sc2_rk_public. The robustness of the resulting clusters for each country was checked by picking 15 clusters at random, plus the top 10 largest clusters, and identifying them on phylogenetic trees and manually verifying that the shared mutations appeared only once, forming a monophyletic cluster.

We grouped the identical sequence clusters into monthly time windows from January to December 2021. We assigned clusters to a given month if at least one sequence was sampled during that month. We focused on this period as the arrival of the SARS-CoV-2 variant of concern (VoC) Alpha in late 2020 led to a scale-up in sequencing efforts in all three countries. Sequencing uptake remained relatively high though 2021 until the arrival of Omicron in late 2021, when a dramatic increase in cases led to a reduction in case ascertainment and sequencing coverage. The number of sequences that were assigned to identical sequence clusters from 1 January 2021 to 31 December 2021 was 96,622 for Switzerland, 267,472 for Denmark, and 355,193 for Germany.

We calculated the sequencing coverage for each month as the ratio of the number of sampled sequences in GISAID over the total number of laboratory-confirmed cases as reported by public health authorities in the respective country. For each country and month we determined among the sequences that have been assigned to identical sequence clusters the number of sequences that have been sampled in the specific month. We directly retrieved daily numbers of newly confirmed cases from public sources of Switzerland [43], Denmark [44] and Germany [45] and summarized them to monthly values. For comparison with our estimates of $R_e$, we also downloaded estimates that were based on laboratory-confirmed cases for all three countries (github.com/covid-19-Re) [7]. To show as additional information in results figures, we retrieved proportions of viral variants among sequences for all three countries on a bi-weekly basis from CoVariants.org [42].

## Mathematical model

We derived a model of the size distribution of identical sequence clusters (Fig 1). To this end, we applied branching process theory to viral transmission, with nodes corresponding to cases and connections between two nodes corresponding to transmission events. We assumed that for each node the number of offspring, i.e., the number of persons to whom an infected individual transmitted the virus, is independent and identically distributed. In addition to the viral transmission process, we consecutively took into account both the mutation of the virus and the incomplete case-detection. At each node of the branching process, a mutation of the virus could occur, with a constant probability for all nodes. Provided a mutation took place, we assumed that it happened prior to the respective node further transmitting the virus or being detected. Consequently, a new mutation was passed on to all secondary cases. This assumption is backed by the work of Braun et al. [46], which showed that overall within-host diversity is low during acute infection and estimated that transmission bottlenecks between hosts are narrow. Furthermore, they argued that the low number of viral particles transmitted while infecting another person can induce a founder effect that wipes out low-frequency intra-host single nucleotide variants. The mutation process allowed us to divide the overall transmission chain into multiple smaller transmission chains by the viral genome sequence present at the nodes. From there we obtained a model of the distribution of the size of identical sequence clusters according to the parameters governing viral transmission and mutation. A detailed description of all different steps can be found in Chapter C.1 of S1 Text.

This approach allowed us to adapt a method previously developed by Blumberg and Lloyd-Smith [34] to estimate $R_e$ and $k$ from the distribution of cluster size to our specific case involving the size of identical sequence clusters. This method assumes a negative binomial distribution for the number of secondary cases with mean $R_e$ (interpreted as the average number of secondary cases per infectious case in the current conditions of transmission and immunity) and dispersion parameter $k$ (lower values implying more dispersion, and thus more super-spreading events). The mean number of mutations per transmission has been estimated to be $M_T = 0.33$ [47]. Based on $M_T$ we can determine the probability $\mu$ that genomes from two subsequent cases in a transmission chain differ from each other by at least one mutation:

$$\mu = 1 - e^{-M_T} . \tag{1}$$

Plugging in $M_T = 0.33$ into the above formula, we get $\mu = 28.1\%$. Alternatively, we estimated $\mu$ from the evolutionary rate $M$ and the serial interval $D$:

$$\mu = 1 - e^{-MD/365.25} . \tag{2}$$

For a within-variant rate $M$ of 14 mutations per year [48] and a mean generation time of $D = 5.2$ days [49], we get $\mu = 18.1\%$. While the former value for $\mu$ might be a slight overestimate since it is not clear whether all transmission pairs used in this estimation are direct transmission pairs, the latter is likely an under-estimate since deleterious mutations contribute more on short time scales within transmission clusters than on longer time scales used to estimate the rate $M$. We use $\mu = 28.1\%$ in the main text and provide an alternative analysis with $\mu = 18.1\%$ in Chapter G of S1 Text.

Since a new mutation splits the transmission chains into identical sequence clusters, we introduced a new metric, the *genomic* reproduction number:

$$R_g = (1 - \mu)R_e \tag{3}$$

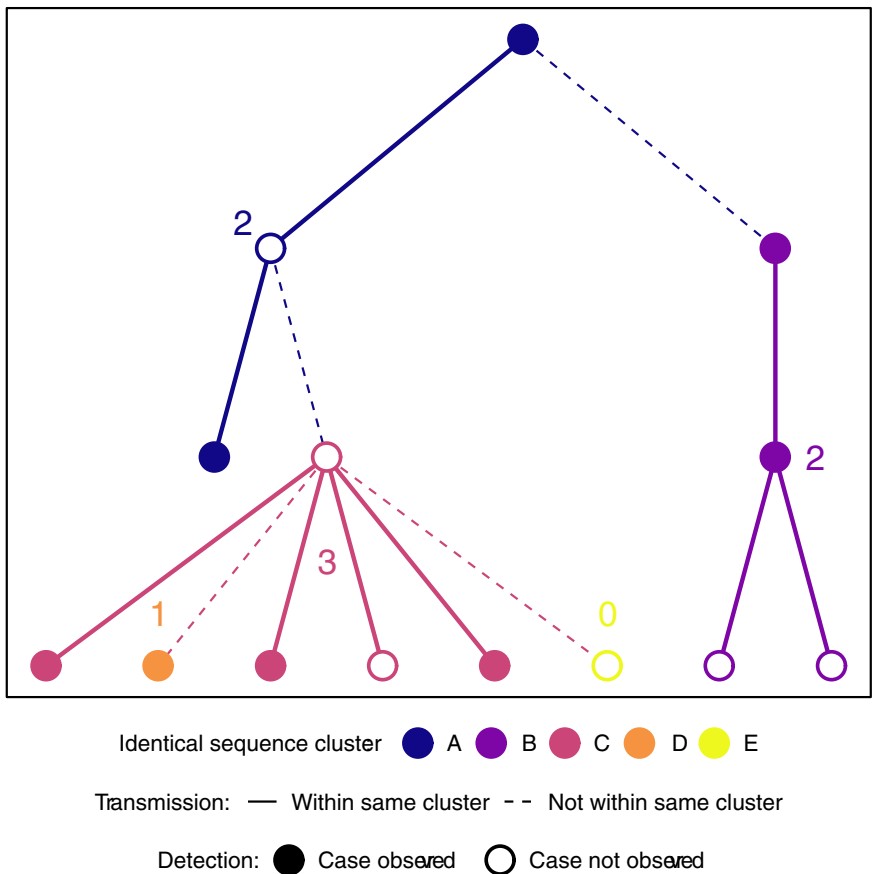

**Fig 1. Transmission chain and identical sequence clusters.** The branching process model creates transmission chains. Viral mutation splits the transmission chains into identical sequence clusters. The numbers indicate the size of the observed identical sequence clusters, i.e., the number of cases in the clusters that were tested and sequenced.

that is related to the effective reproduction number of cases $R_e$ and corresponds to the average number of secondary cases within a given identical sequence cluster. Since the number of secondary cases follows a negative binomial distribution with mean $R_e$ and dispersion parameter $k$, the number of secondary cases that belong to the same identical sequence cluster as their source case also follows a negative binomial distribution with mean $R_g$ and dispersion parameter $k$. We have included a complete mathematical derivation of this property in Chapter B.1 of S1 Text.

A key assumption of the model is that $R_g \leq 1$. If $R_g > 1$, then a case in an identical sequence cluster creates on average more than one secondary case that belongs to the same cluster. This results in a non-zero probability that such an identical sequence cluster will not go extinct if allowed to grow over a long time period. This introduces practical issues for inference that are beyond the scope of this work. We present our ideas on how to extend our model to deal with a scenario in which $R_g > 1$ in Chapter C.3 of S1 Text.

As a last step, we incorporated incomplete case-detection assuming an independent observation process [38]. We computed the probability that a case was detected as the product of the probability that a case was confirmed by a test and the probability that the viral genome of a confirmed case was sequenced.

### Bayesian inference

We developed a Bayesian inference framework to estimate the model parameters from the size distribution of identical sequence clusters and implemented in `Stan`, a probabilistic modeling platform using Hamiltonian Monte Carlo methods [50]. We used weakly-informative prior distributions for $R_e$, $k$, and the testing probability and, depending on the manner the mutation process was included into the model, a weakly-informative prior distribution for the mutation probability $\mu$ or an informative prior distribution for the yearly mutation rate $M$. The mean generation time $D$ and the sequencing probability were taken as fixed values. We estimated $R_e$, $k$, and the testing probability for monthly time windows from January to December 2021. While the main focus was to estimate $R_e$ and $k$, inclusion of the testing probability allows for our approach to be generalizable to contexts where few data exist about this quantity. In addition, this allows incorporating the uncertainty about testing in the final estimates of $R_e$ and $k$. The posterior samples were summarized by their mean and 95% credible interval. More details can be found in Chapters D and F of S1 Text. The R and `Stan` code files are provided within the R package `estRodis`, available on the following GitHub repository: github.com/mwohlfender/estRodis.

### Simulations

We conducted a simulation study to validate the model. We generated data of identical sequence clusters for different parameter combinations of $R_e$, $k$, and the mutation, testing and sequencing probability. To this end, we performed stochastic simulations of transmission using branching processes and then applied mutation and incomplete case-detection. Cases with an identical sequence were grouped in a cluster. We subsequently applied our inference framework to the simulated data and compared the mean and the 95% credible interval of the generated samples of the posterior distributions to the true values (Fig 2). In addition, we computed the coefficient of variation and the root mean square error between the mean of the posterior distribution of the estimated parameters and the true values (Figs B and C in S1 Text) and the coverage of the true value by the 95% credible interval of the posterior distribution (Fig D in S1 Text).

### Results

Based on the new metric of the *genomic* reproduction number $R_g$, we developed a Bayesian inference model to estimate $R_e$, $k$, and the testing probability from data on the size distribution of identical sequence clusters. We validated this model using simulated data and investigated how combinations of $R_e$, $k$, the testing probability, and the sequencing probability could be accurately recovered. To this end, we ran 10 rounds of simulation of 3,000 identical sequence clusters for 300 parameter combinations (Fig 2). Generally, we found that the error between the true and estimated values of $R_e$ and $k$ is minimized for higher testing and sequencing probabilities. Our method provided reliable estimates of $R_e$ and $k$ as long as $R_e$ was below 1.4 (i.e., $R_g$ is below 1 given our assumption for the mutation rate, Fig 2A and 2B). When $R_e$ was set to 1.5 (so that $R_g$ is above 1), the inference model could not reliably recover the true value anymore.

Several patterns are recognizable in the results of the simulation study. First, the estimates of $R_e$ tend to increase when the testing probability gets bigger. This effect is stronger in situations with a smaller underlying true effective number. Second, the higher the testing or the sequencing probability, the more precisely the dispersion parameter is estimated. Furthermore, in some scenarios there is not enough information in the data for the model to move

away from the prior distribution of the dispersion parameter, which has mean 0.5. This is particularly evident in situations with lower underlying true values of the effective reproduction number, the testing or the sequencing probability. Overall, the validation showed that our model can be used to estimate $R_e$ and $k$ in specific settings, especially in situations with high testing and sequencing probabilities, but fails to provide a reliable estimate of the testing probability.

We identified clusters of identical SARS-CoV-2 sequences using real-world sequencing data from Switzerland, Denmark, and Germany. From a total of 96,622 sequences from Switzerland, 267,472 from Denmark, and 355,193 from Germany, we identified 58,587, 84,537 and 218,497 clusters of identical sequences, respectively. Small clusters dominated the distribution of cluster size, with 79.7%, 70.8%, and 79.4% of the clusters being of size one in Switzerland, Denmark, and Germany, respectively. The mean cluster size was 1.65 in Switzerland, 3.16 in Denmark, and 1.63 in Germany, which suggests a higher probability of testing and/or sequencing in Denmark compared to the other countries. The sequencing coverage for laboratory-confirmed SARS-CoV-2 cases indeed differed substantially between the three countries over time, with an average sequencing coverage of 10.4% for Switzerland, 38.2% for Denmark, and 6.4% for Germany over 2021.

Each identical sequence cluster was assigned to a given month if it contained at least one sequence from that month, thus we assigned 3603 (6.1%) identical sequence clusters from Switzerland, 6610 (7.8%) from Denmark and 14,520 (6.6%) from Germany to at least two months of 2021. Including the identical sequence clusters overlapping from 2020 into 2021 or from 2021 into 2022, the number of identical sequence clusters extending beyond one month among the identified clusters was 4151 (7.1%) in Switzerland, 8144 (9.6%) in Denmark and 15,992 (7.3%) in Germany. The small proportions of clusters extending beyond one month support the assumption that $R_g \leq 1$ in these particular settings, justifying the choice of a monthly time window.

The number of clusters assigned to a month and their mean size varied by country as well as by month (Fig 3). More details about the size distribution of identical sequence clusters of Switzerland, Denmark or Germany assigned to a month of 2021 are presented in Chapter A of S1 Text.

We applied the model to the size distribution of identical sequence clusters to obtain monthly estimates of $R_e$, $k$, and the testing probability for SARS-CoV-2 in Switzerland, Denmark, and Germany (Fig 4). The estimated $R_e$ values per month for each country across 2021 did fluctuate through the year (Fig 4A). The average $R_e$ value for Switzerland, Denmark, and Germany was 1.02 (95% credible interval, CI: 0.84-1.22), 1.18 (95% CI: 0.97-1.40), and 1.06 (95% CI: 0.89-1.24), respectively. Due to the relatively large time window of one month, the underlying trends in $R_e$ as estimated based on laboratory-confirmed cases were not captured consistently using our new method based on identical sequence clusters, notably at time points with transitions between SARS-CoV-2 variants.

Our estimates of the dispersion parameter $k$ varied between the different countries, with average estimates over the 12 month period of 0.17 (95% CI: 0.02-0.30) for Switzerland, 0.38 (95% CI: 0.13-0.60) for Denmark, and 0.15 (95% CI: 0.02-0.27) for Germany. Estimates of $k$ were highest in Denmark, fluctuating around 0.3 to 0.5 from January to October before dropping at the end of the year. Depending on the country and month, the estimates of $k$ have narrower or wider credible intervals. In Switzerland and Germany, the estimates of $k$ were generally lower with values around 0.1 to 0.3 and also dropped at the end of the year. The transition from Alpha to Delta was not associated with a substantial change in the estimates of $k$.

The estimated testing probability was relatively high and came with more uncertainty. This indicates that the testing probability cannot be precisely estimated from the size distribution

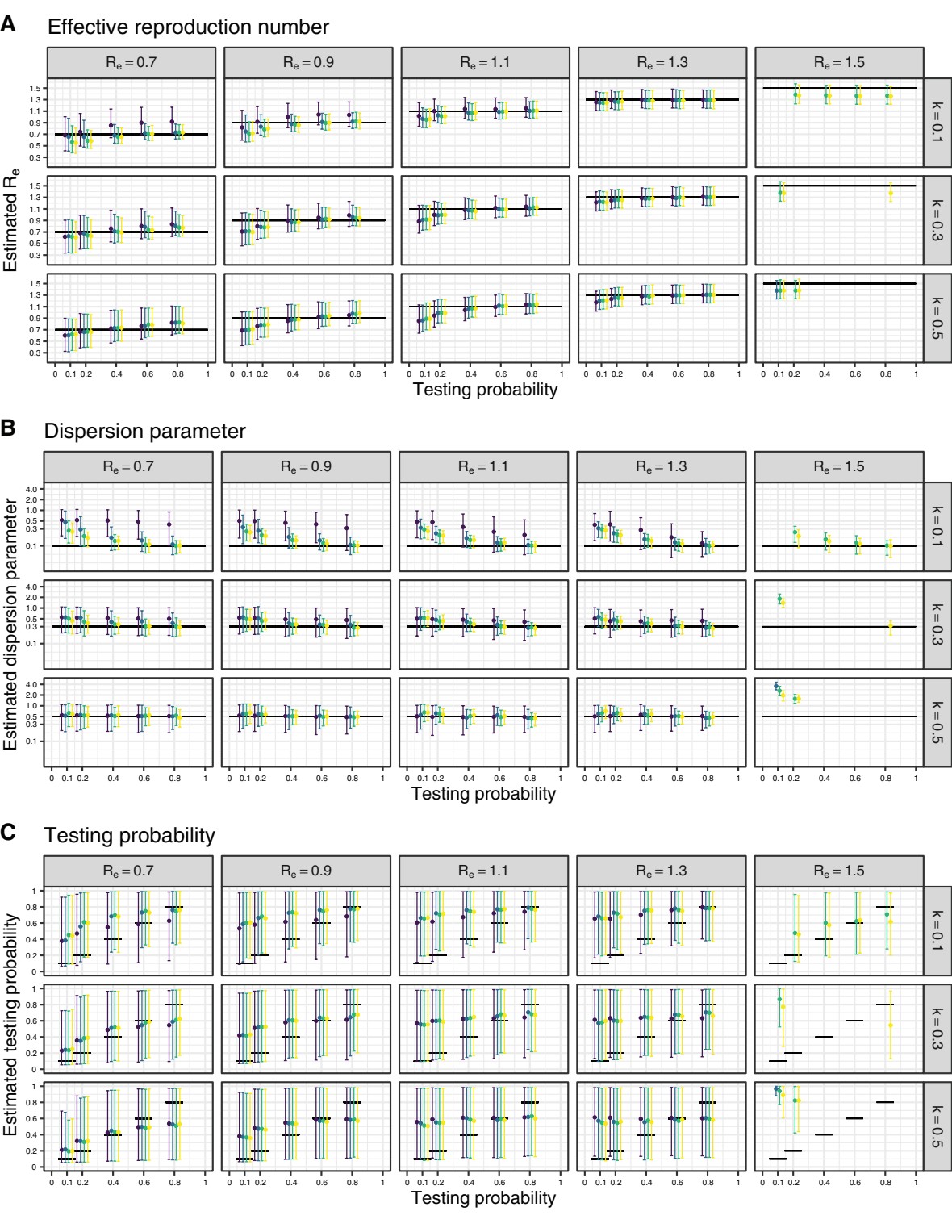

**Fig 2. Validation of the Bayesian inference model to estimate $R_e$, $k$ and $\tau_{test}$ from the size distribution of identical sequence clusters.** (**A**) Estimate of the effective reproduction number $R_e$. (**B**) Estimate of the dispersion parameter $k$. (**C**) Estimate of the testing probability $\tau_{test}$. True values are shown as black lines. For each parameter combination, we ran the model 10 times on 3,000 simulated clusters each. The generated samples of the posterior distributions are summarized by mean and 95% credible interval.

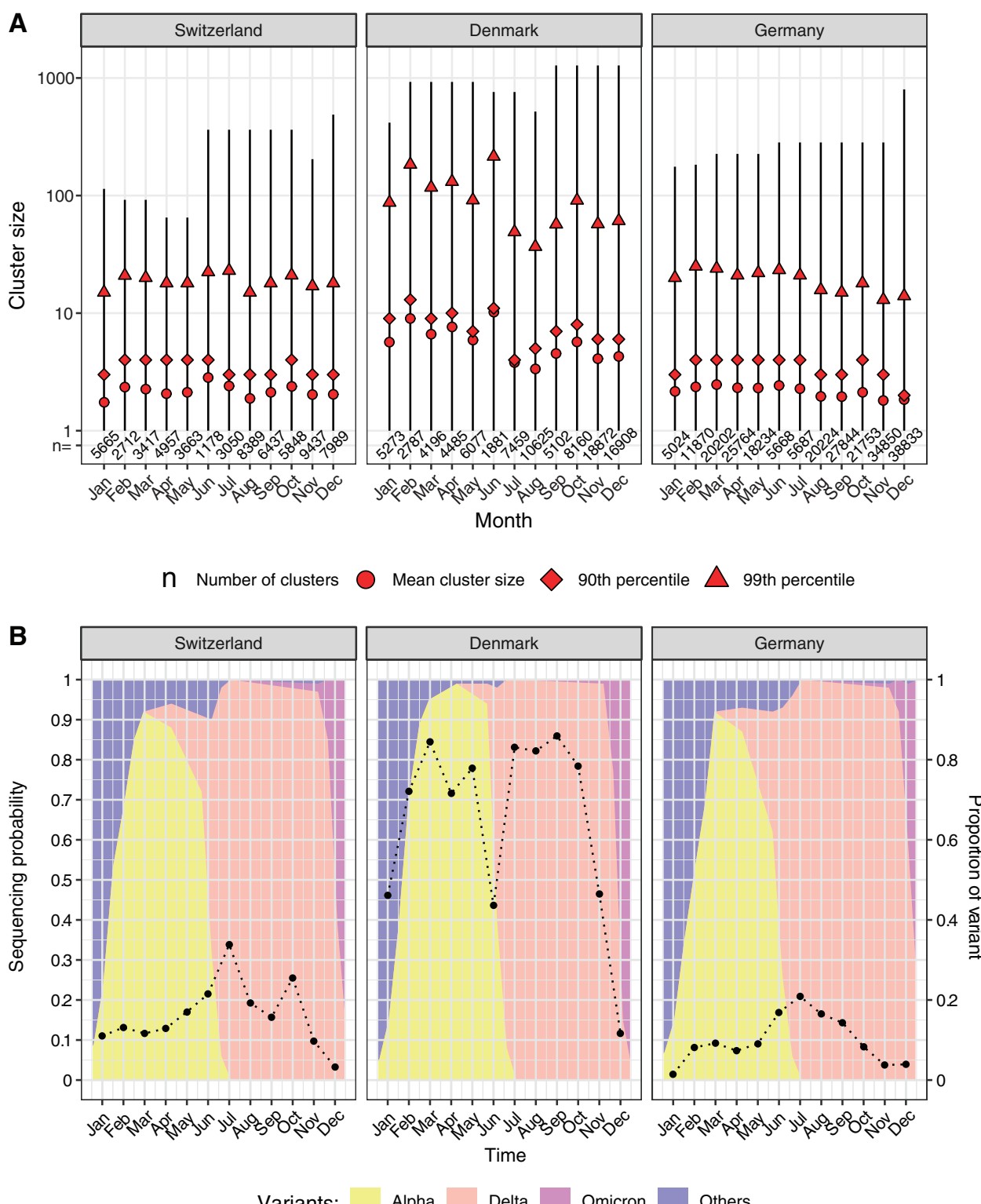

**Fig 3. Identical sequence cluster size distribution and sequencing coverage for SARS-CoV-2 in Switzerland, Denmark, and Germany in 2021.**
(**A**) Range, mean, 90th percentile and 99th percentile of the identical sequence cluster size distribution and number of clusters by month based on data from GISAID. (**B**) Sequencing coverage (dots) and proportion of SARS-CoV-2 variants (background color) by month.

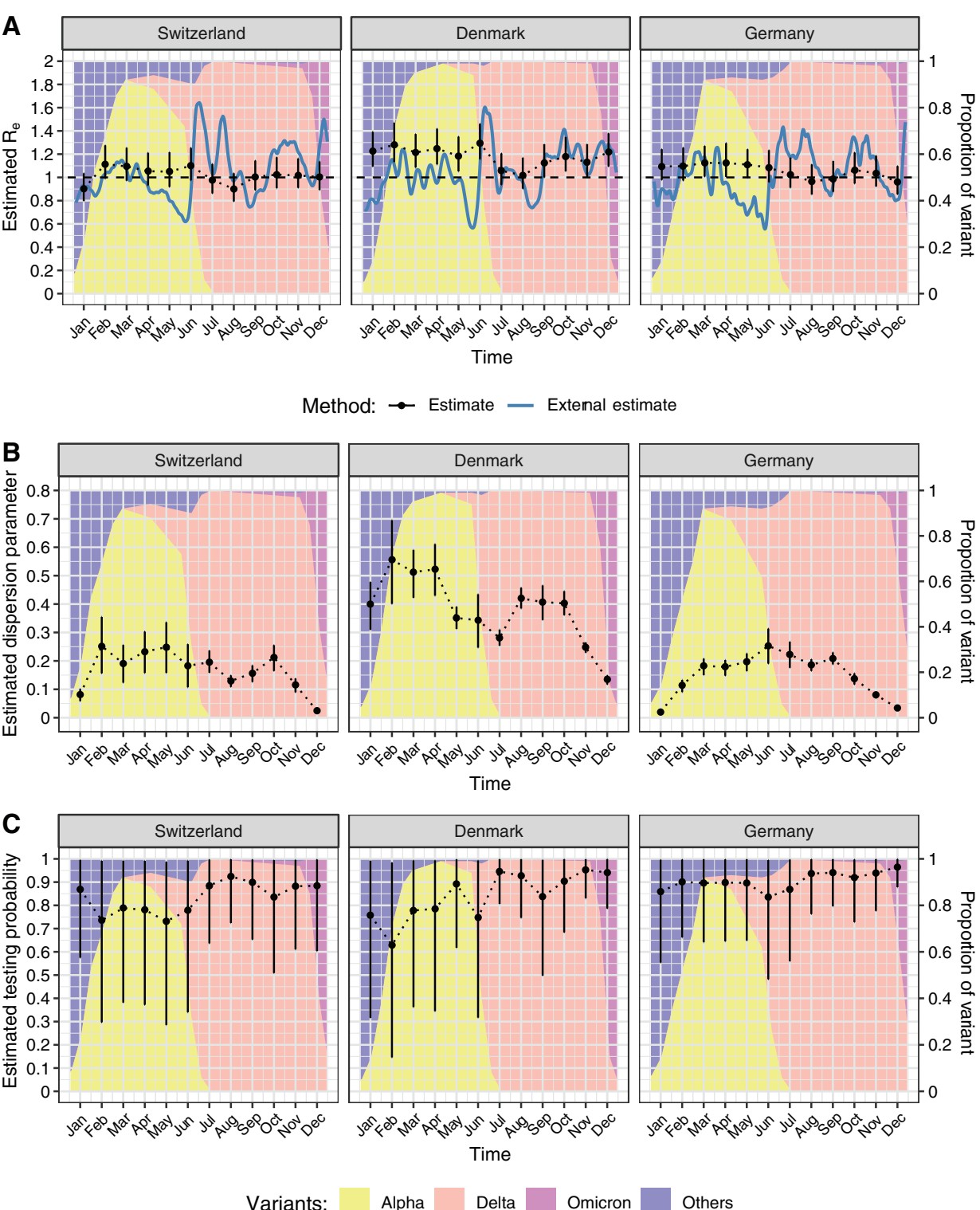

**Fig 4. Parameter estimates based on the size distribution of identical SARS-CoV-2 sequence clusters in Switzerland, Denmark and Germany in 2021.** A: Effective reproduction number $R_e$. B: Dispersion parameter $k$. C: Testing probability. The estimates are based on monthly time windows of identical sequence clusters. For each month the estimated mean and 95% credible interval of the posterior distribution (in black) are shown. $R_e$ values are compared to external estimates based on laboratory-confirmed cases (in blue, from github.com/covid-19-Re [7]).

of identical sequence clusters using our Bayesian inference model. Thus, we conducted a sensitivity analysis where the testing probability was fixed at 57.6% from January to August 2021 and at 35% from September to December 2021, based on estimates of the case ascertainment using swab positivity in England [51]. This did impact $R_e$ and $k$ estimates to a small degree, generally increasing $R_e$ estimates slightly and decreasing $k$ estimates slightly (Chapter G of S1 Text).

Furthermore, we have carried out a posterior predictive check and a goodness of fit analysis to assess the compatibility of our simulation of identical sequence clusters based on the estimated parameters with the cluster data from Switzerland, Denmark and Germany (Chapters I and J of S1 Text).

## Discussion

We developed a novel Bayesian model to estimate the effective reproduction number, $R_e$, and crucially the level of superspreading of an epidemic from viral sequence data, $k$. To this end, we introduced a new metric, the *genomic* reproduction number, $R_g$, the mean number of secondary cases created by an infected individual that share the same viral genome. $R_g$ is therefore the analogue of $R_e$ in the context of clusters of identical sequences. We validated our model on simulated data and applied it to the size distribution of identical sequence clusters of SARS-CoV-2 in Switzerland, Denmark, and Germany in 2021. We obtained monthly estimates of $R_e$ around or below the epidemic threshold of 1. The estimates of the dispersion parameter $k$ varied substantially by country and month and were typically between 0.1 and 0.5. Together, our study illustrates how the increasing amount of viral sequence data can be used to inform epidemiological parameters of SARS-CoV-2 and potentially other pathogens.

During our work, Tran-Kiem and Bedford (2024) [39] published a study based on the same underlying theory as ours. The fact that two research groups applied the same concepts at the same time independently of each other to estimate parameters related to transmission dynamics of infectious diseases is certainly linked to the increasing availability of viral genomic data, and underlines the promising nature of this approach. A major difference between the two studies is the choice of the method for parameter inference. Instead of a maximum likelihood approach, we have decided to use a Bayesian framework allowing the full propagation of the uncertainty related to data and incompletely known input parameters throughout the estimation process. Furthermore, while Tran-Kiem and Bedford (2024) [39] applied their model to rather small datasets of different diseases where traditional phylodynamic methods would also have been applicable, we have dealt with significantly larger country-wide datasets covering the whole year 2021. We have developed approaches to efficiently identify clusters of identical sequences within these datasets containing hundreds of thousands of sequences and applied our model to monthly segments of the data to estimate the effective reproduction number $R_e$ and the dispersion parameter $k$ and track their change over the year 2021. These points demonstrate the different applications of the underlying theory and the complementary nature of these two studies.

The main strength of our study is the presentation of a new method to estimate epidemiological parameters, specifically the overdispersion in the number of secondary cases, from viral sequence data alone. While $R_e$ can often be reliably estimated from counts of laboratory-confirmed cases as well, estimating $k$ has historically been considerably more challenging since it requires direct information about transmission events, for example, the number of secondary cases created by individual cases or the size of transmission chains. With the novel

method presented here, this challenge can be overcome through making use of the large number of sequences available for SARS-CoV-2. In addition, our method uses a computationally efficient way to identify clusters of identical sequences without the need to reconstruct very large phylogenies. Finally, the Bayesian inference model allows a full propagation of the uncertainty in the viral transmission parameters, the mutation rate, and the incomplete case-detection.

Our study comes with a number of limitations. First, the presented method requires a large amount of sequence data and a relatively high case ascertainment to obtain reliable estimates of epidemiological parameters. We expect such datasets to become more common with further technological developments in diagnostics and sequencing, in particular for outbreaks and epidemics of emerging infectious diseases. Second, miscalls in sequences, such as reversions to a reference sequence, might have resulted in some clusters that are artificially separated or combined. We countered this problem to a certain extent by replacing variant-defining sequences with the variant name. For the remaining sequences, we expect that miscalls would have only little impact on the overall cluster size distribution and the parameter estimates. Third, to limit computational complexity of the model we assumed an independent observation process, i.e., infected cases are detected randomly and tested cases are sequenced at random. It is possible that this assumption is not perfectly realistic, which could contribute to the large credible intervals for the estimates of the testing probability. For example, testing and sequencing uptake might cluster in certain settings, such as during contact tracing and outbreak investigations. This would on average lead to a higher actual detection probability for cases contained in a large identical sequence cluster and to a lower actual detection probability for cases contained in a small identical sequence cluster. Therefore, compared to the true underlying size distribution of identical sequence clusters, smaller identical sequence clusters would rather be underrepresented and larger clusters rather overrepresented in the observed data. To find the combination of $R_e$, $k$, mutation and testing probability best fitting the data, our model would then be biased to higher values for $R_e$ compared to the true underlying value. In our model of the size distribution of identical sequence clusters, a lower value for $k$ increases the variance and consequently leads to both a higher probability of identical sequence clusters of size one and a higher probability of large identical sequence clusters. We expect that this property limits the impact of a deviation of the observed data from the true underlying distribution on the estimation of $k$. A less frequent occurrence of smaller clusters points towards a larger $k$, a more frequent occurrence of larger clusters to a smaller $k$. In short, we think that the effects on the estimations of $R_e$ and $k$ resulting from this are limited. Lastly, we did not consider changes in the mutation rate and the generation time during the study period, which could have led to different ratios between $R_e$ and $R_g$, especially for different SARS-CoV-2 variants.

The simulation study showed that our model does not provide equally precise parameter estimates in the entire space of possible values for $R_e$, $k$ and the testing probability. We consider the main reason for this phenomenon to be that in some scenarios, especially when the testing probability and the sequencing probability are low, only a very small number of different sizes of identical sequence clusters appears in the data, in extreme cases only two. The probabilities of larger cluster sizes can become very small very quickly with increasing cluster size, which implies a large amount of data would be required to observe some of them. The more large clusters there are in the data, the more likely the model considers a larger value of $R_e$. This explains the tendency of increasing estimates of $R_e$ when the testing probability increases in the simulation study. What is also relevant here is our choice of the prior distribution of $k$, a gamma distribution with shape 5 and rate 10, i.e., with mean 0.5. If the model is not able to retrieve information from the data to update the prior distribution, it will stay with

it. This is a potential reason why the model fails to provide a precise estimate of $k$ in particular when the underlying true $k$ is small. Since for each month there are identical sequence clusters of at least 28 different sizes in Switzerland, 94 in Denmark and 52 in Germany contained in the dataset, we expect that there is sufficient information available to the model to come up with solid estimates of $k$ when applied to the real-world data.

During some months, our estimates of $R_e$ differ notably from estimates of $R_e$ derived from laboratory-confirmed cases and were not able to fully capture the changes observed in the comparator estimate. On the one hand, this could simply be due to the use of two different methods with different assumptions [52]. On the other hand, this could be due to the assumption of a fixed time window, resulting in monthly rather than daily estimates of $R_e$. Due to this rather low time resolution during an ongoing epidemic, rapid increases and decreases in $R_e$ cannot be observed and are averaged into a mean estimate for the corresponding time period. As the estimates of $R_e$ based on laboratory-confirmed cases can be obtained on a daily basis, this more traditional method is more suited to detect sudden changes of $R_e$.

The allocation of the identical sequence clusters to monthly subsets introduces potential biases due to left and right censoring. On the one hand, we partly circumvented this problem by assigning identical sequence clusters to a given month if at least one sequence was sampled during that month. On the other hand, this has the disadvantage that we included clusters in our analysis that can span much longer time periods than one month and that cover different epidemic phases with widely different values of $R_e$. Furthermore, the transition periods from one SARS-CoV-2 variant to another pose an additional challenge to our inference method, as by definition sequence clusters cannot span over these transitions. Thus, it is not surprising that our method cannot capture the rapid changes of $R_e$ and its rise after the arrival of new variants, such as the growth of Delta and Omicron in June and December 2021, respectively. The issue with left and right censoring also means that our method cannot reliably estimate $R_e$ when the genomic reproduction number $R_g$ is above 1, i.e., when some clusters can grow - in theory - indefinitely. We considered ways to enable the model to deal with situations in which $R_g$ is larger than 1, but we have not found a suitable solution. We set out our considerations in Chapter C.3 of S1 Text. Together, these limitations can explain why our estimates of $R_e$ seemed to fluctuate around the epidemic threshold of 1, which corresponds to the long-term average of an ongoing epidemic. For further exploration of the impact of different model assumptions on $R_e$ estimates, see Chapter G of S1 Text.

In contrast to the increasingly precise estimates of $R_e$ based on laboratory-confirmed cases, estimates of the dispersion parameter $k$ for SARS-CoV-2 have typically come with considerable uncertainty within studies and substantial variability between studies [18–32]. Furthermore, Zhang et al. [53] and Ko et al. [54] showed substantial changes of $k$ over time. Our monthly estimates of $k$ also vary considerably between countries and over time (range: 0.15-0.38), but fit well within the range of earlier estimates. Therefore, our framework appears better suited for estimating $k$ than $R_e$, as it performs comparably to existing methods for estimating $k$, while more precise methods are available for estimating $R_e$ using laboratory-confirmed case data.

Interestingly, $k$ was estimated to be considerably higher in Denmark compared to Switzerland and Germany, indicating fewer superspreading events. While Switzerland and Germany had similar levels of testing uptake and laboratory-confirmed cases during most of 2021, the testing uptake in Denmark was considerably higher while the number of laboratory-confirmed cases was similar to the other countries. This suggests a higher case ascertainment and a better control of SARS-CoV-2 transmission in Denmark, which could have led to fewer opportunities for superspreading events and, as a consequence, higher values of $k$. Based on the same underlying theory, Tran-Kiem and Bedford [39] estimated $k$ at 0.63 (95% confidence

interval: 0.34-1.56) in New Zealand, which is somewhat higher than our estimates. Again, this can be explained by the fact that they analyzed identical sequence clusters during a period when there was a high level of transmission control with little opportunities for superspreading events. Lastly, it remains unclear how our estimates of $k$ were affected by the left and right censoring of clusters, but the relatively low estimates in January and December 2021 could be impacted by the early spread of Alpha and Omicron. Another potential explanation is that superspreading events may be favored by indoor gathering during winter periods.

Another potential source of bias of the estimates of both $R_e$ and $k$ is the import and export of cases. An imported case from abroad most likely carries a mutation not yet present in the country and hence starts a new cluster of identical sequences. Imported cases could on one hand create fewer secondary cases compared to local cases because their infectious period already might have started prior to arrival. Tsang et al. [55] took this into account in their study by adjusting the infectiousness profiles of imported cases. Furthermore, the imposition of quarantine measures was likely effective in reducing transmission [56]. On the other hand, Creswell et al. [57] argue that frequent travelers might have more contacts than those individuals who do not travel abroad. This would lead to a higher risk of transmission. An exported case leads to the corresponding identical sequence cluster not getting as big as it would in a scenario without exportation of cases. Therefore, in general the importation and exportation of cases could increase the overall number of identical sequence clusters and reduce the mean size of identical sequence clusters. Such a deformation of the size distribution of identical sequence clusters could lead to an underestimation of both the effective reproduction number and the dispersion parameter. However, we expect that the proportion of imported, respectively exported, cases in the dataset is small. Therefore, we assume that our results are not significantly influenced by case importation and exportation.

The vast amount of genomic data that was generated during the pandemic provides novel opportunities to characterize and track the transmission dynamics of SARS-CoV-2. Our newly developed Bayesian model to infer epidemiological parameters from the size distribution of identical sequence clusters can reliably estimate the level of superspreading through the dispersion parameter $k$ and has the potential to inform about $R_e$, albeit with some practical limitations for the latter. With the increasing affordability and ease of sequencing, we expect that large volumes of sequence data will become more readily available in the future, making it possible to adapt this method to other pathogens and to estimate the transmission heterogeneity in different countries, for different variants, and over time.

## Acknowledgments

We gratefully acknowledge all data contributors, i.e., the authors and their originating laboratories responsible for obtaining the specimens, and their submitting laboratories for generating the genetic sequence and metadata and sharing via the GISAID Initiative, on which this research is based. We also gratefully acknowledge and thank all labs around the world that have collected and shared SARS-CoV-2 sequences we used in our study, and in particular the labs in Switzerland, Germany, and Denmark, where the sequence data for this study originated. A complete list of the labs that generated the data we used from GISAID can be found in the EPI_SETs. We additionally thank Pierre-Yves Boëlle for valuable feedback on the project and for the suggestion to validate the compatibility of the simulation of identical sequences with cluster data from Switzerland, Denmark and Germany by a posterior predictive check. We also thank Eugenio Valdano for a stimulating discussion of the project and his advice on how to assign clusters of identical sequences to months. Calculations were performed on UBELIX (www.id.unibe.ch/hpc), the HPC cluster at the University of Bern.

## Supporting information

**S1 Text. Additional results, figures and tables supporting the main text.**
(PDF)

**S1 Table. GISAID EPI_SET Table 1—EPI_ISL identifiers for sequences from Switzerland.**
(PDF)

**S2 Table. GISAID EPI_SET Table 2—EPI_ISL identifiers for sequences from Denmark.**
(PDF)

**S3 Table. GISAID EPI_SET Table 3—EPI_ISL identifiers for sequences from Germany.**
(PDF)

## Author contributions

**Conceptualization:** Emma B. Hodcroft, Christian L. Althaus.

**Data curation:** Emma B. Hodcroft.

**Formal analysis:** Emma B. Hodcroft, Martin S. Wohlfender.

**Investigation:** Emma B. Hodcroft, Martin S. Wohlfender, Julien Riou, Christian L. Althaus.

**Methodology:** Emma B. Hodcroft, Martin S. Wohlfender, Julien Riou, Christian L. Althaus.

**Software:** Emma B. Hodcroft, Martin S. Wohlfender.

**Supervision:** Richard A. Neher, Christian L. Althaus.

**Validation:** Martin S. Wohlfender, Richard A. Neher, Julien Riou.

**Visualization:** Martin S. Wohlfender.

**Writing – original draft:** Emma B. Hodcroft, Martin S. Wohlfender, Christian L. Althaus.

**Writing – review & editing:** Emma B. Hodcroft, Martin S. Wohlfender, Richard A. Neher, Julien Riou, Christian L. Althaus.

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
