## [Decision Letter · Decision Letter 0]

29 Jul 2024

Dear Dr Hodcroft,

Thank you very much for submitting your manuscript "Estimating Re and overdispersion in secondary cases from the size of identical sequence clusters of SARS-CoV-2" for consideration at PLOS Computational Biology.

As with all papers reviewed by the journal, your manuscript was reviewed by members of the editorial board and by several independent reviewers. In light of the reviews (below this email), we would like to invite the resubmission of a significantly-revised version that takes into account the reviewers' comments.

Two of the three Reviewers have raised a point about the results presented in this manuscript being too close or identical to the work published in a recent paper of Tran-Kiem and Bedford (https://www.pnas.org/doi/10.1073/pnas.2305299121 ). It it essential that the authors clearly indicate both in the revised version of the manuscript, and in their responses the novelty of their method and/or results in relation to that work. The Reviewers have also made a number of other suggestions on how the improve the paper and make the presentation clearer, so I suggest the authors carefully go through them and address them.

We cannot make any decision about publication until we have seen the revised manuscript and your response to the reviewers' comments. Your revised manuscript is also likely to be sent to reviewers for further evaluation.

Sincerely,

Konstantin B. Blyuss

Academic Editor

PLOS Computational Biology

Hannah Clapham

Section Editor

PLOS Computational Biology

Two of the three Reviewers have raised a point about the results presented in this manuscript being too close or identical to the work published in a recent paper of Tran-Kiem and Bedford (https://www.pnas.org/doi/10.1073/pnas.2305299121 ). It it essential that the authors clearly indicate both in the revised version of the manuscript, and in their responses the novelty of their method and/or results in relation to that work. The Reviewers have also made a number of other suggestions on how the improve the paper and make the presentation clearer, so I suggest the authors carefully go through them and address them.

Reviewer's Responses to Questions

**Comments to the Authors:**

Reviewer #1: Please see attachment

Reviewer #2: I think this is a great, useful idea, that is well-explained. I don't have much in the way of criticisms or suggestions.

Some small points:

-From text on page 5, and also in Section 3, it was not 100% clear to me that when a mutation occurs at a node, then it is assumed that the node *has* that mutation. This is clear from the figure on page 5. There is an assumption built in here which is that any mutation occurs early in the life of the pathogen within one host, both before anyone else is infected and also before the host is sampled. I think this is a good assumption, but you certainly could imagine situations where it is violated.

- As the authors acknowledge, the error bars on the estimates of R_e are suspiciously small. Part of the issue is that the model assumes things that aren't true (like R_e being constant over the course of a month). But I wondered about D and \tau_sequence being fixed parameters. We know that D can change with conditions in a pandemic, and even if it didn't, I don't think we know it with high precision.

-In section 4 of the SI, I didn't understand the table entry for \tau_sequence "1.42 − 85.94%". Also, I was curious about the scaled beta for \tau_test. How did expert knowledge lead to a distribution where it's sharply peaked above 0.05 but zero below it? It may be some Bayesian thing I don't understand.

-page 4. "To minimize the chance... only sequences with more than four mutations were used." I would have appreciated "four mutations with respect to the reference genome" or something like that.

Even smaller points:

page 3, line 48. The first two sentences of the paragraph start with the same phrase. Also, is the pandemic still ongoing?

page 8, fig 2, thin out the y axis labels so they are not overlapping?

page 12, line 320, "events may *be* favoured" ?

Reviewer #3: In this manuscript, Hodcroft and colleagues propose to use sets of identical viral sequences (readily available from public databases) to estimate the time-varying effective reproduction number R_e and, more critically, the overdispersion parameter k of the negative binomial distribution that quantifies the extent of transmission heterogeneity. The overall idea of the manuscript is interesting, as case data alone are not very informative of the extent of transmission heterogeneity, and studies of detailed outbreaks are few. As such, approaches such as the one proposed show great promise to getting a handle on the extent of viral superspreading that occurs and how this changes over time. I do, however, have some major concerns with the manuscript as is and I hope that my recommendations below prove helpful.

Major concerns/suggestions:

1. The authors briefly mention the published work of Tran-Kiem and Bedford (PNAS) on lines 59-60. How does the method in that published paper compare to the current method? The authors indicate that this is a related study using the same approach. In the discussion, they then say it is a similar approach. Is it exactly the same approach, so if the authors used their approach on the same dataset from New Zealand (line 313), would they estimate the same k of 0.63, or how do the approaches differ? More detailed information on this would be very helpful, especially to gauge the novelty of the currently submitted work.

2. Equation (1) for mu: instead of using the estimated substitution rate (M) and the generation interval (D set to 5.2 days) to estimate mu = 0.181, I recommend using the quantified probability of mu from established transmission pairs. This is a more relevant data point, given that estimates of M are impacted by purifying selection (Ghafari et al. MBE 2022) and the generation interval similarly can change substantially over time because of control efforts (see, e.g., Ali et al. Science 2020) or because of viral differences (e.g., generation interval differences by VOC). Park et al. Nature Communications (2023) provide an empirical estimate of the mean number of mutations that occur at transmission of mu = 0.33 (95% CI = 0.22-0.48; their Fig 6C) from transmission pair data. This is the mean of a fit Poisson distribution. Therefore, the probability of 0 mutations at transmission is given by exp(-mu) = 0.7189, which is ballpark similar to Hodcroft et al.’s (1-0.181) = 0.8190, but the CI based on the empirical transmission pairs yields a range between 0.6188 to 0.8025, which does not include 0.8190.

3. The authors indicate that if the number of offspring is negative binomially distributed with mean R_e and overdispersion parameter k, then the number of viral offspring that are genetically identical to the focal individual is similarly negative binomially distributed with mean R_e(1-mu) and overdispersion parameter k. By the same reasoning, this would indicate that the number of viral offspring that are genetically dissimilar to the focal individual would be distributed with mean R_e(mu) and overdispersion parameter k. If this is the case, is the sum of two random variables drawn from these distributions negative binomially distributed with mean R_e and overdispersion parameter k?

4. Supplemental methods: Section 2: probability theory: the structure of this could be improved upon, at least by adding an introduction as to why the specific lemmas and theorems are being provided to the reader. Also (obviously minor): Table S7A: ]…[ should be […].

5. Figure 1: I was initially confused by this figure, because the two solid blue colored nodes do not need to be a transmission pair. There could be multiple individuals between them that are all unsampled. Perhaps instead show a transmission tree, highlighting sampled individuals?

6. Figure 2: This figure is meant to show how the method does on simulated data, however, it’s very difficult to interpret the results from these RMSE plots, and I highly recommend revising the figure. For example, could you plot estimated versus true R_e, estimated versus true k, estimated versus true testing probability in some way where the readers could see how close the estimates are to their true values and if estimates are biased in one direction of the other? From Figure 2 I have a very hard time interpreting whether this method does a good job or a terrible job.

7. In the supercritical case (R_e > 1), the same expression holds (see Blumberg and Lloyd-Smith (2013) Epidemics section 2.3.1). Given that this is the case, could you then fit k in the supercritical case of R_e > 1 by just looking at the sizes of the subset of identical clusters, conditioning on their extinction?

8. Is the underlying distribution generated under the estimated R_e and k model consistent with the observed distribution of cluster sizes (at some instance in time)? Visualization of goodness of fit between the model (parameterized by fitting to cluster size data) and the cluster size data would be helpful.

**Have the authors made all data and (if applicable) computational code underlying the findings in their manuscript fully available?**

Reviewer #1: Yes

Reviewer #2: Yes

Reviewer #3: None

PLOS authors have the option to publish the peer review history of their article (what does this mean?). If published, this will include your full peer review and any attached files.

Reviewer #1: No

Reviewer #2: No

Reviewer #3: No
---

## [Decision Letter · Decision Letter 1]

6 Dec 2024

PCOMPBIOL-D-24-00848R1

Estimating Re and overdispersion in secondary cases from the size of identical sequence clusters of SARS-CoV-2

PLOS Computational Biology

Dear Dr. Hodcroft,

Thank you for submitting your manuscript to PLOS Computational Biology. After careful consideration, we feel that it has merit but does not fully meet PLOS Computational Biology's publication criteria as it currently stands. Therefore, we invite you to submit a revised version of the manuscript that addresses the points raised during the review process.

Please submit your revised manuscript within 60 days Feb 05 2025 11:59PM. If you will need more time than this to complete your revisions, please reply to this message or contact the journal office at ploscompbiol@plos.org. Please include the following items when submitting your revised manuscript:

We look forward to receiving your revised manuscript.

Kind regards,

Konstantin B. Blyuss

Academic Editor

PLOS Computational Biology

Hannah Clapham

Section Editor

PLOS Computational Biology

Feilim Mac Gabhann

Editor-in-Chief

PLOS Computational Biology

Jason Papin

Editor-in-Chief

PLOS Computational Biology

**Reviewers' comments:**

Reviewer's Responses to Questions

**Comments to the Authors:**

Reviewer #1: I appreciate the thoughtful set of revisions that the authors have made based on the reviewer comments. I do have some lingering concerns though.

I still feel that the manuscript oversells the novelty as compared to the published work of Tran-Kiem and Bedford. If the key distinction is that inference is done in a Bayesian rather than a frequentist framework, then I think this should be made explicit throughtout – including in the abstract. (E.g. in the abstract, instead of saying ‘we developed a mathematical model for the distribution of the size of identical sequence clusters, in which we integrated viral transmission, the mutation rate of the virus, and incomplete case-detection’, I would say something like ‘We adapted a model … for Bayesian inference’)

Here is some feedback regarding my three ‘major’ concerns (replicated below, along with follow-up feedback)

1. '… the inference method is predicated on the assumption that R_g is less than ...'

I appreciate the additional analysis that the authors did regarding the proportion of clusters that did not end within a month. Within the text, I would add an explicit explanation for why this proportion is important (i.e. explicitly mention the assumption that the genomic reproduction number is less than one, and that the validity of this assumption can be assessed by evaluating the proportion of clusters that did not end within a month).

2. 'In figure 4 and S3 there are notable time periods where the R estimates from sequences and case counts differ markedly. ….'

I appreciate the extra sensitivity analyses the authors have done regarding R_e, but the variability of R estimates from case counts compared to sequences is still notable (e.g.R of 1.6 from case counts vs 1.1 from sequence data). It would help to learn about potential reasons why there is so much volatility in the case base estimates compared to the genomic estimates.

3. '…the estimates of the dispersion parameter are assumed to be more reliable than the estimates of R_e….'

I am confused by the explanation provided. To my mind any change in transmission has the capability of influencing both R and k. For example, decrease in R due to immunity is likely going to be patchy (and thus contribute to heterogeneity). Likewise changes in R due to various public health interventions aren’t expected to be uniformly applicable across individuals. In fact, some studies have looked at temporal trend of the dispersion parameter and found it can vary more than R (e.g. https://doi.org/10.1371/journal.pcbi.1010078). I recognize that relating the time dependence of R and k is probably beyond the scope of this paper, so I would just be more careful with the wording. I.e. I wouldn’t take for granted that k doesn’t change. I would rather explain what modeling assumptions this corresponds to and ideally find data to support it (or run some sensitivity analyses to evaluate how important the assumption is).

Lastly, regarding minor concerns, the one small thing I saw is that I would edit the following, ‘For example, a lower probability of observing an identical sequence cluster of size one would indicate a larger value of k, whereas a higher probability of observing large clusters would indicate a lower value of k’. I would make it clear whether you are talking about the true or inferred value of k. It’s also a little confusing since these scenarios seem similar – i.e. if there is a relatively lower probability of observing isolated clusters, then it seems that there would be a relatively higher probability of observing large clusters?

Reviewer #2: Thanks for the changes.

Reviewer #3: I thank Hodcroft and coauthors for their thorough response to my initial concerns and suggestions. I think the figures are very much improved over the original ones and that the text is more clearly presented. I especially think that the change in the text relating to the per transmission per genome mutation rate M_T has clarified the work.

I have four remaining comments/concerns, two of which are major and two of which are minor:

1. (MAJOR) Related to my original concern (3):

Lines 170-173 in the main text still state: Since the number of secondary cases follows a negative binomial distribution with mean Re and dispersion parameter k, the number of secondary cases that belong to the same identical sequence cluster as their source case also follows a negative binomial distribution with mean Rg and dispersion parameter k.

This statement is also restated in Supplemental Material section 2.

As far as I can tell, this statement is not true, and this is what I was getting at in my original concern (3). If this statement were true, then the number of secondary cases that do not belong to the same identical sequence cluster as their source case would also follow a negative binomial distribution with mean mu*R_E and dispersion parameter k. But if these statements were true, then the sum of random variables from these distributions should follow a negative binomial distribution with mean R_E and dispersion parameter k, but this is not the case (as the authors agree, given their initial response to my comment). So, it stands that the statement in lines 170-173 is not true. I don’t want to stand as a gatekeeper to this manuscript being published, but it really seems to me that if I am correct with this (and I’m happy to be proven wrong), then the whole approach falls apart. If this is the case, one could say, “we make an approximation that Rg follows a negative binomial distribution with mean Rg and dispersion parameter k. This approximation (detail how/under what parameterizations it does best and worst)…”

2. (MINOR) Related to my original concern (4). ][ vs [] in Table S7A. I thank the authors for clarification! I have never seen this notation. For support that does not include the endpoints, I have always seen “(.., …)” rather than “]…, …[ “). Maybe this notation differs across geographic regions. This reviewer is from the US.

3. (MINOR) In Figure 2: Label columns with R_E (rather than R), to make clear that these simulations were run with these effective reproduction numbers (not genomic reproduction numbers of this magnitude)

4. (MAJOR) Figure 2: currently, the results on the simulated data are interpreted very superficially, in lines 215-217 of the text (“Generally, we found that the error between the true and estimated values of Re and k is minimized for higher testing and sequencing probabilities. Our method provided reliable estimates of Re and k as long as Re was below 1.4.”) The benefit of testing an approach on simulated data is that biases and other patterns can be detected! There are several other notable patterns that need to be mentioned and interpreted. First, as testing probability increases, Re estimates tend to increase. This is particularly the case at low R_E. Why is this the case? Second, and this is really important given what the approach presented here is trying to do: estimates of the dispersion parameter k are biased high at low testing probabilities when k < 0.5. Why is this the case? I believe the reason for this is related to my remaining concern (1) above, namely that the assumption that R_g is distributed with mean (1-mu)*R_e and same dispersion parameter k, when the mean is indeed (1-mu)*R_e but the dispersion parameter should no longer be k. Third, testing probability is generally not identifiable. All of these patterns/biases should be explicitly mentioned and interpreted in the main text such that the reader can appreciate the limitations of the approach.

**Have the authors made all data and (if applicable) computational code underlying the findings in their manuscript fully available?**

Reviewer #1: Yes

Reviewer #2: Yes

Reviewer #3: Yes

PLOS authors have the option to publish the peer review history of their article (what does this mean?). If published, this will include your full peer review and any attached files.

Reviewer #1: No

Reviewer #2: No

Reviewer #3: No

**Figure resubmission:**
---

## [Decision Letter · Decision Letter 2]

11 Mar 2025

Dear Dr Hodcroft,

We are pleased to inform you that your manuscript 'Estimating Re and overdispersion in secondary cases from the size of identical sequence clusters of SARS-CoV-2' has been provisionally accepted for publication in PLOS Computational Biology.

Best regards,

Konstantin B. Blyuss

Academic Editor

PLOS Computational Biology

Hannah Clapham

Section Editor

PLOS Computational Biology

Reviewer's Responses to Questions

**Comments to the Authors:**

Reviewer #1: Thank you for your thoughtful responses to reviewer feedback, and great work overall.

Reviewer #3: Thank you for such a thorough response to my remaining questions/concerns.

**Have the authors made all data and (if applicable) computational code underlying the findings in their manuscript fully available?**

Reviewer #1: Yes

Reviewer #3: None

PLOS authors have the option to publish the peer review history of their article (what does this mean?). If published, this will include your full peer review and any attached files.

Reviewer #1: **Yes: **Seth Blumberg

Reviewer #3: No

---

## [Editor Report · Acceptance letter]

PCOMPBIOL-D-24-00848R2

Estimating Re and overdispersion in secondary cases from the size of identical sequence clusters of SARS-CoV-2

Dear Dr Hodcroft,

I am pleased to inform you that your manuscript has been formally accepted for publication in PLOS Computational Biology. Your manuscript is now with our production department and you will be notified of the publication date in due course.

With kind regards,

Olena Szabo
